# Nanomedicine as a Potential Tool against Monkeypox

**DOI:** 10.3390/vaccines11020428

**Published:** 2023-02-13

**Authors:** Nura Adam Mohamed, Luisa Zupin, Sarah Ismail Mazi, Hebah A. Al-Khatib, Sergio Crovella

**Affiliations:** 1Biomedical Research Center (BRC), Qatar University, Doha P.O. Box 2713, Qatar; 2Institute for Maternal and Child Health IRCCS Burlo Garofolo, 34137 Trieste, Italy; 3Department of Cardiac Sciences, College of Medicine, King Saud University, P.O. Box 7805, Riyadh 11472, Saudi Arabia; 4Biological Science Program, Department of Biological and Environmental Sciences, College of Arts and Sciences, Qatar University, Doha P.O. Box 2713, Qatar

**Keywords:** monkeypox, nanomedicine, vaccine

## Abstract

Human monkeypox is a rare viral zoonosis that was first identified in 1970; since then, this infectious disease has been marked as endemic in central and western Africa. The disease has always been considered rare and self-limiting; however, recent worldwide reports of several cases suggest otherwise. Especially with monkeypox being recognized as the most important orthopoxvirus infection in humans in the smallpox post-eradication era, its spread across the globe marks a new epidemic. Currently, there is no proven treatment for human monkeypox, and questions about the necessity of developing a vaccine persist. Notably, if we are to take lessons from the COVID-19 pandemic, developing a nanomedicine-based preventative strategy might be prudent, particularly with the rapid growth of the use of nanotechnology and nanomaterials in medical research. Unfortunately, the collected data in this area is limited, dispersed, and often incomplete. Therefore, this review aims to trace all reported nanomedicine approaches made in the monkeypox area and to suggest possible directions that could be further investigated to develop a counteractive strategy against emerging and existing viruses that could diminish this epidemic and prevent it from becoming a potential pandemic, especially with the world still recovering from the COVID-19 pandemic.

## 1. Introduction

Monkeypox is an infectious disease caused by the monkeypox virus (MPXV), a double-stranded DNA virus belonging to the Orthopoxvirus genus (subfamily Chordopoxvirinae, family Poxviridae) [1,2]. In addition to the monkeypox virus, 11 different species of orthopoxvirus have been identified, including the camelpox virus, cowpox virus, horsepox virus, and variola virus. The vairola virus, which can cause smallpox disease, is the best-known species of this genus. It caused several epidemics throughout history before being completely eradicated in 1980 [3]. The disease was successfully eradicated by using a closely related virus, the vaccinia virus, as a smallpox vaccine [2]. Following the eradication of smallpox, monkeypox has become the most significant orthopoxvirus concern for health authorities due to occasional outbreaks in endemic countries. The clinical features of monkeypox disease are similar to those observed in smallpox-affected patients. However, the symptoms are much less severe, and in most cases, the outcome is complete recovery. Smallpox vaccines have been shown to confer immunity against the MPXV. However, the discontinuation of the national vaccine campaign made the younger population susceptible to the virus [1].

The MPXV was first isolated from *Cynomolgus macaques* while conducting research on polio vaccines in 1958 [4]. In 1970, a smallpox-like disease was reported in a 9-month-old boy in the Democratic Republic of the Congo, where it became endemic [5]. Since 1970, MPX outbreaks have been reported in 11 African countries, including Cameroon, Nigeria, the Central African Republic, and Gabon. The number of cases increased dramatically in endemic African countries during 2010–2019, reaching 18,000 suspected and confirmed cases. In 2003, the virus was reported for the first time in the USA, resulting in 70 cases. After that, monkeypox outbreaks outside of Africa were frequently identified in travelers from Nigeria to the United Kingdom (2018–2022), Singapore (2019), and the USA (2021) [1]. In May 2022, multiple cases of monkeypox were reported in several non-endemic countries, such as the USA, UK, France, Brazil, and others. As of 20 January 2023, the World Health Organization reported 84,916 cases and 81 deaths across 110 countries, showing a lower mortality rate than the previous outbreaks (Figure 1).

The genetic classification of the MPXV categorizes the virus into two clades: the central African clade and the west African clade. The former is endemic in Congo Basin countries, while the latter is endemic in West African countries. In addition to the distinct geographical distribution, the two clades differ in disease severity and transmissibility. The Congo Basin clade has been associated with more severe symptoms, higher transmissibility, and an increased case fatality rate of 1–10%.

Phylogenetic analysis of the currently circulating MPXV suggests that the outbreak virus belongs to the less severe West African clade [13]. Moreover, the analysis has revealed a close genetic similarity with the MPXV isolated during the 2018–2019 outbreaks in the UK and Singapore [14]. Similar to other DNA viruses, poxviruses have a relatively low mutation rate (10^−^^5^ to 10^−6^ mutations per replication site). Surprisingly, though, viruses isolated during the current outbreak exhibited more modifications than expected. On average, 50 single-nucleotide mutations were identified compared to the 2018–2019 virus [15,16]. Yet, the impact of these mutations on virulence and transmissibility needs to be investigated [9,17].

## 2. Viral Replication Cycle

The MPXV is an enveloped, oval-shaped virus measuring 200–250 nm in size. The enveloped virus encloses a linear, double-stranded DNA genome and enzymes required for virus replication [18]. The MPXV genome is around 200 Kb, which encodes 190 genes. The genes for replication, transcription, and assembly are localized in the highly conserved core region. In contrast, the genes involved in host range and pathogenicity are mapped at the genome’s left and right variable arms, where the inverted terminal repeats are present, forming hairpin loops [19]. Most of the virologic data about poxviruses are derived from the prototype vaccinia virus (VACV). A poxvirus replication cycle generates two types of particles: the mature virion (MV), which consists of one lipid bilayer, and the extracellular virion (EV), which contains a second external membrane. MVs are the most abundant particles released by cell lysis, mediating the transmission between hosts. On the other hand, EVs are generated by Golgi apparatuses or endosomes. They remain associated with the host cellular membrane until they are expelled, leading to local transmission within the host [20]. Viral proteins that recognize host cell receptors have not yet been identified, but based on studies of the VACV, different host cell proteins have been discovered. MVs attach to the extracellular glycoprotein laminin through the viral protein A26 [21]. Other viral surface proteins have also been identified, including D8, which binds glycosaminoglycans, and A27 and H3, which bind to heparan [22]. For EVs, no target factor has been identified.

The main internalization route occurs through macropinocytosis for both MVs and EVs, although direct fusion with the plasma membrane is also documented for EVs [22]. Macropinocytosis requires multiple cell factors and needs epidermal growth factor receptor signaling. However, MVs depend on phosphatidylserine (PS), while EVs do not rely on PS exposure [23]. After membrane penetration, endogenous RNA polymerase and transcription factors start early viral gene expression. Then the high-packed core releases its content into the cytoplasm, where the viral DNA functions as a template for genome replication and subsequent intermediate and late viral gene expression [24,25]. Translation of viral proteins and genome replication form the intracellular mature virus, which migrates to the Golgi to form the intracellular enveloped virus. Cell-associated enveloped viruses are formed when virions fuse with cellular membranes. By rearrangement of the cellular actin tail, they are released into the extracellular environment or delivered as free extracellular vesicles (Figure 2) [26].

## 3. Tropism, Transmission, Symptoms, and Disease Course

A study on an immune-deficient mouse model demonstrated diverse tissue tropisms for the MPXV, including in the brain, heart, kidney, ovary, liver, pancreas, and lung [27]. However, a histopathologic study on monkeys determined that the MPXV’s primary target is the lymphoid tissue [28]. Several cell lines (e.g., Vero, HeLa, and A549) and laboratory animals (e.g., mice, rabbits, rats, and non-human primates) are susceptible to MPX infection [28]. So far, no wildlife reservoir has been identified for the MPXV. However, the MPXV has been isolated from the Funisciurus anerythrus squirrel [29], the Cercocebusatys monkey [30], rodents, Gambian rats, and other primates [31].

The MPXV is primarily considered to be a zoonotic viral infection. Transmission has occurred almost exclusively from animals to humans through bites, eating undercooked meat, or direct contact with biological fluids from infected animals [32]. Before 2000, most cases were of zoonotic origin and were commonly reported in young children (with an average age of 4.4 years) [33]. In the current outbreak, human-to-human transmission has become the main route of transmission, with sexual transmission being the primary mode of diffusion [13], in addition to long-term close contact (respiratory droplets), direct contact with body fluids or contaminated objects, as well as vertical transmission from infected mothers [34]. Approximately 98.2% of cases occur among men who have sexual contact with men, homosexuals, and bisexuals. Based on recent epidemiological data collected during the current outbreak, 60% of sexually transmitted cases occurred at parties. Moreover, most confirmed cases are also immunodeficiency virus (HIV)-positive [13].

The MPX infection has an incubation period of 1–2 weeks, followed by the prodromal phase lasting 0–2 days, and the rash stage persisting for 1–3 weeks. The disease begins with general symptoms, such as fever, fatigue, headache, muscular ache, and lymphadenopathy (Figure 3). Then, after 1–5 days, the rash appears, and the patients become contagious. The progression of skin lesions occurs in this order: plaques, papules, blisters, pustules, scabs, and shedding. Rashes are localized in the conjunctiva (20%), oral mucosa (70%), palms and soles (75%), face (95%), and genitals (30%) [4]. It is commonly a self-limited disease with complete recovery after a few weeks. However, complications may also occur, especially among children and young adults who did not receive the smallpox vaccine. Complications include conjunctivitis, keratitis, skin infections, dehydration, encephalitis, and pneumonia [9]. Moreover, the severity and fatality rate differ between the two clades, occurring in only 1% of the West African clade [35] and up to 10% for the Congo Basin clade [36] (Figure 3).

## 4. Diagnostic Tools, Treatment, and Preventative Strategies

Several methods have been developed for detecting the MPXV nucleic acid, among which real-time PCR (RT-PCR) is the preferred routine method (Figure 3). Typically, the following genes are selected as RT-PCR targets: (i) B6R (the envelope protein gene); (ii) E9L (the DNA polymerase gene) [37,38]; (iii) RPO18 (the DNA-dependent RNA polymerase subunit 18 gene) [39]; (iv) the complement binding protein C3L [40]; and (v) F3L and N3R genes [41]. Notably, the specificity and sensitivity of available RT-PCR kits are yet to be improved. Therefore, whole genome sequencing of the virus genome is considered the gold standard method for distinguishing the MPXV clades and the MPXV from other orthopoxviruses [42,43]. Despite being accurate, sequencing technologies are costly ($15–20 per sample), making them inaccessible in many countries. Therefore, cheaper and more available techniques were used, such as (i) the restriction-length fragment polymorphism, (ii) loop-mediated isothermal amplification technology [44], and (iii) the recombinase polymerase amplification [45].

An enzyme-linked immunosorbent assay (ELISA) is another detection method that could be used to detect MPX infection. This technique is cheaper than RT-PCR and genome sequencing, costing around $6–8 per sample. An ELISA can detect the presence of specific IgM and IgG antibodies in the serum of the MPX patients at days 5–8 after the infection. Collected data showed a 4-fold increase in patients’ serum antibodies during the acute and convalescent stages of the MPX infection. However, due to the cross-reaction between the MPX viral antigens and the other poxviruses, the specificity of this technique is insufficient, making it inaccurate in identifying the MPXV [18]. To assist the MPX diagnosis, electron microscopy can be used for the morphological characterization of the virus. However, this method can only provide clues that the virus belongs to the poxvirus family because it cannot distinguish between the morphologies of the MPXV and other poxviruses. Furthermore, the sample preparation procedure is highly complex and complicated, making it impractical for routine use, as well as time-consuming and costly, and the electron microscope has a low sensitivity. Other diagnostic tests include immunochemistry analysis and multiplexed immunofluorescence imaging that could be used to detect the MPX viral antigen [46], as well as viral isolation and culture from the patient’s specimen for viral characterization and establishing definitive diagnostic tools [4].

The MPXV is a re-emerging threat to humanity that has gained increasing global attention in recent years. As of January 2023, 84,916 confirmed human cases had been reported globally [47]. Even though no licensed antiviral drugs are currently available to treat a human MPX viral infection, clinical management of the MPXV should be adjusted accordingly. This could help reduce symptoms, manage complications, and minimize long-term consequences. In Europe, the European Medical Association (EMA) has recently licensed the use of Tecovirimat, a small molecule virus inhibitor, for MPX treatment. Experimental data collected from Tecovirimat treatment in human and animal studies showed safe and tolerable outcomes with only minor side effects [47]. Tecovirimat is administered orally or intravenously and is used for adults and children weighing at least 3 kg [48]. Other drugs have also been found to confer anti-poxvirus activity in both in vitro and in vivo experiments. These treatments include Cidofovir (a viral DNA polymerase inhibitor), Brincidofovir (a prodrug of Cidofovir), and vaccinia immunoglobulin [47,48]. Cidofovir is given either intravenously or topically, and Brincidofovir is given orally [48]. However, no evidence has been found yet to support the efficacy of Brincidofovir and Cidofovir in treating MPX [47].

Similarly, the effectiveness of vaccinia immunoglobulin, which is given intravenously, in human subjects still needs to be better established [47]. Furthermore, pregnant women with severe disease are given either tecovirimat or vaccinia immunoglobulin, as they are preferred over Brincidofovir and Cidofovir, particularly with Cidofovir being a teratogen [49] (Figure 1). Furthermore, the MPXV is often associated with bacterial infections, which should be treated to avoid any complications that might result from an MPX viral infection, according to clinical practice guidelines [1].

To face the outbreak, new MPX vaccines are being developed. Meanwhile, the USA has approved vaccines that were proven effective in eradicating smallpox. Observational studies demonstrated that smallpox vaccination prevents MPX 89% of the time [1,47,50,51]. As a result, people who have been vaccinated against smallpox may have milder illnesses during an MPX infection [1,47,50,51]. The immune response provides cross-protection against orthopoxviruses, and this cross-protection is used to develop smallpox and MPX vaccines that are based on the vaccinia virus [1,47,50,51]. In 2019, a novel two-dose MPX vaccine was developed using a modified attenuated vaccinia virus (Ankara strain). Unfortunately, the vaccine is still in short supply. Another vaccine, ACAM2000, was approved by the FDA for use in adults (>18 years old) at high risk of contracting smallpox. The vaccine, which is formulated from a live vaccinia virus (replication-competent, 2nd generation vaccine), was shown to protect against the MPXV and hence can be used as an alternative to the JYNNEOS™ vaccine [1,47,50,51]. The JYNNEOS™ vaccine, a non-replicating vaccinia virus (3rd generation vaccine), also known as Imvamune or Imvanex, was approved for use against MPX and smallpox in the USA. Preclinical data on animal models have demonstrated good immunogenicity and effectiveness against the MPXV, favoring its approval [1,47,50,51]. Both JYNNEOS™ and ACAM2000 are based on live viruses propagated in cell cultures.

ACAM200, based on a live replication-competent virus, replicates and produces a mild infection once in the host, inducing an immune response similar to the natural infection. Nevertheless, in immunocompromised patients, infants, pregnant or breastfeeding women, and individuals with a weak immune system, there is a residual risk of a severe or even fatal infection, and the vaccine is contraindicated. Moreover, an increased risk of adverse effects may occur in individuals presenting cardiac disease (risk for myopericardititis) or dermatologic conditions (risk for skin infection, erythema, and eczema) [52,53]. JYNNEOS™, based on a non-replicating virus, presents a better safety profile and less severe adverse effects [52]. To date, the use of JYNNEOS™ or ACAM2000 is not recommended for children, as their administration in the pediatric age group is considered off-label and has been suggested only for postexposure prophylaxis in emergencies [52].

ACAM2000 is supplied as a lyophilized powder, whereas JYNNEOS™ is in a frozen liquid form. However, both vaccines require storage at −20 °C before utilization, requiring cold chain monitoring to maintain their characteristics [52]. The global distribution of these vaccines presents an unequal allocation, with African countries the most disadvantaged, mainly due to their pricing, storage, and administration by healthcare professionals [54,55]. As previously described, smallpox vaccines confer immunity against the MPXV; however, global orthopoxvirus immunity has decreased since the eradication of the smallpox virus and the cessation of compulsory smallpox vaccination in many countries between 1970 and 1980. This has resulted in an increase in MPX cases, particularly among unvaccinated children, adolescents, and young individuals.

Moreover, the containment and eradication efforts will be challenging due to the spread of the virus globally and the circulation of the virus in both human and non-human reservoirs. Lessons learned from the ongoing COVID-19 pandemic have emphasized the importance of developing preventative and therapeutic measures to avoid any catastrophic outcomes of the current MPX outbreak [48]. Several approaches are being considered for developing new vaccines against the MPXV, including the rapidly growing nanotechnology field.

## 5. Utilizing Nanomedicine Applications to Face Monkeypox

Despite the progress in developing conventional vaccines for the orthopoxvirus, improvements are still required, especially with the low immunogenicity, toxicity, instability, and multiple dose concerns associated with the conventionally available vaccines that currently employ a live attenuated vaccinia virus and not the MPXV itself or derived antigens. Furthermore, viral infections pose significant global health challenges, especially with the emergence of more resistant, rapidly evolving viruses causing new pandemics and epidemics, thus emphasizing the potential of nanomedicine applications in the field of antiviral therapies. In addition to providing delivery tools, nanomedicine can increase cellular and humoral immune responses. The involvement of nanoparticles in vaccine development can enhance the immunogenicity and stability of antigens and modulate the immune response. Therefore, over the past decades, nanoparticles such as polymers, liposomes, virus-like particles, immunostimulant complexes, inorganic nanoparticles, and emulsions have been designed and studied to stabilize vaccine antigens and act as adjuvants (Figure 4).

Despite having many different nanoformulation-based vaccines, such as the virus-like particle, viral vector vaccines, and nucleic acid vaccines that are encapsulated in lipid-based nanoparticles, which are used in clinics for other viral diseases, as well as the others, including organic and inorganic nanomaterials, so far there are no nano-based vaccines in clinical use for the MPXV [56,57,58,59,60]. However, of the studied nanoparticles, metal-based nanoparticles have a vital role in presenting their activity without incorporating drugs. Numerous activities of metal-based nanoparticles are already well documented, such as blocking the host-virus interaction of the MPXV, competitive inhibition of the herpes simplex virus, preventing the viral binding of the hepatitis B virus, inactivating the tacaribe virus, and retarding viral attachment with the glycoprotein of the HIV-1 virus [56]. Furthermore, metal nanoparticles, such as iron oxide nanoparticles (FeNPs), are known to have antiviral activity, acting as viral reservoirs and chelating the virus circulating in the bloodstream, which could be used in designing detection tools for the MPXV [57]. Additionally, FeNPs can interfere with the different stages of the viral life cycle and modulate the immune response, as well as selectively targeting lung endothelial cells, liver cells, and spleen cells. Similar properties seen in other viruses can be tested to determine if FeNPs will have a similar effect on the MPXV. This is important as the MPXV largely affects the lung, liver, and spleen, and FeNPs are known to accumulate in these organs in addition to their anti-inflammatory activity [57,58,59,60]. Of the nanoparticle’s properties, the immunomodulatory activity is attributable to its small size, which enhances its recognition and facilitates its uptake by the antigen-presenting cells. Furthermore, the surface functionalization of nanoparticles using different moieties permits the targeted delivery of antigens to the desired cells with specific receptors, thereby stimulating selective and specific immune responses [58].

In the present scenario, nanoparticles (especially metals) and their unique chemical and physical properties are emerging as novel antiviral agents [61]. This is simply because, compared to conventional antivirals, metals can attack different virus targets, lowering the possibility of developing resistance. Moreover, metal nanoparticles have been studied for their antiviral potential and antibacterial activity against Gram-negative and Gram-positive bacteria [61]. Additionally, when designing nanovaccines and nanotherapeutic options for viral infections, it is important to understand the viral life cycle, as metal nanoparticles were shown to prevent or interfere with one or more stages of the viral life cycle [61]. This can facilitate the design of nanoagents that can interfere with one of the crucial viral life stages, such as preventing its entry into the host cell, the genome replication and assembly, the production of mature virions, and its release from the host cell. Theoretically, the antiviral activity of any metal can be tested; however, more investigations need to be made in that area.

Possible metal nanoparticles employed for this purpose include copper, zinc, titanium, magnesium, gold, alginate, and silver nanoparticles against the hepatitis B virus, respiratory syncytial virus, influenza virus, HIV-1, herpes simplex virus type 1, MPXV, and Tacaribe virus [61]. Interestingly, gold nanoparticles (AuNPs) were shown to have inhibitory effects on the viral infectivity and spread of the MPXV [60]. The uniqueness of the AuNPs depends on their resistance to tarnishing. Earlier records date the use of gold for medical purposes back to the Chinese civilization in 2500 BC. Since then, scientists have suggested utilizing Au-based nanomaterials to treat viral diseases, including smallpox, skin ulcers, measles, and syphilis [62]. Another nanoparticle that gained attention in the field of viral vaccine development is silver nanoparticles (AgNPs), with results showing the nanoparticles’ antiviral activity against the MPXV, hepatitis B virus, HIV, herpes simplex virus, and respiratory syncytial virus. Furthermore, other studies showed that small AgNPs (10 nm) showed a higher antiviral potency and were more effective in reducing plaque formation against the MPXV compared to larger sizes (20–80 nm) [63]. The AgNPs might exert such an effect by intervening during the early steps of viral binding and penetration into the host cell. Another theory is that once the virus enters the host cell, silver nanoparticles disrupt the intracellular pathways important for virus replication [64]. The interaction of nanoparticles with microorganisms is a developing area of research that has included evaluating the antimicrobial capacity of certain silver-containing nanoparticles against vegetative bacteria and HIV-1 [5,6]. Previous research on the nanoparticle-HIV-1 interaction demonstrated that silver-containing nanoparticles inhibited HIV-1 infectivity in vitro by binding to the disulfide bond regions of the CD4 binding domain within the gp120 glycoprotein subunit. The binding of these nanoparticles to the gp120 subunit appeared to be size-dependent, as particles larger than 10 nm were not observed attached to the viral envelope [65]. Based on this previous study, using silver-containing nanoparticles as an antiviral therapeutic agent may be a new area for developing nanotechnology-based antiviral therapeutics for the MPXV [65].

In terms of organic nanoparticles (e.g., lipid), a research group developed lipid nanoparticle (LNP)-formulated mRNA vaccines and investigated their effect on enhancing the immune response both in vitro and in vivo for use in MPXV prevention. The results showed the mRNA constructs’ translation, secretion, and biological activation, as evidenced by the increase in the target monoclonal antibody levels (c7D11, c8A, and c6C). These data demonstrated the feasibility of inducing multiple antibodies through mRNA constructs using nanotechnology [66]. Another study by Alec et al. developed an mRNA vaccine that encodes for four highly conserved MPXV surface proteins. These proteins are involved in the attachment, entry, transmission, and induction of MPX immunity. The vaccine was conjugated with an LNP, generating an mRNA-LNP vaccine. The results showed the superiority of the mRNA-lipid nanoparticle vaccine in neutralizing and spreading the inhibitory activity against the MPXV in tested animals compared to the lethal vaccinia virus (VACV)-treated groups. Their remarkable discovery is promising and could be a game-changer that needs further investigation [67]. In another study, Wang et al. employed nanoparticles in devising a diagnostic test for the MPX viral infection and the differentiation between the West and Central African MPXV isolates. The nanoparticle-based biosensor detection tool was named MPXV-MCDA-LFB, and in this test, there were two sets of multiple cross displacement amplification (MCDA) primers: D41L was designed to target the Central African MPXV, and ATI was designed to target the West African MPXV. The reaction briefly works by conducting an isothermal MCDA reaction for the DNA templates followed by the lateral flow biosensor detection of the preamplification target sequences, with the optimal reaction temperature and time being 64 °C for 30 min. The detection tool developed in this study was shown to be effective and rapid [68]. Moreover, the efficient inhibition of cell-pathogen interactions to prevent infections is an urgent yet unsolved problem. However, in their study, Benjamini et al. managed to develop functionalized multivalent 2D carbon nanosystems to investigate their antiviral effects. In vitro investigations that determined the ability of the orthopoxvirus strains to enter cells (Vero E6 (ECACC 85020206) and Hep2 (ECACC 86030501) showed that the 2D carbon nanosystems had excellent binding and efficient inhibition of orthopoxvirus infections. Their results suggest that these nanosystems are promising candidates to help us develop potent virus inhibitors [69]. Finally, a study conducted by Solenne et al. suggested inhibiting orthopoxvirus infections by blocking viral replication by using a small interfering RNA to target the D5R gene. Their study showed the potential of siRNA in treating different poxvirus infections, including smallpox and MPX. However, this will require using a carrier to ensure the safe delivery of the siRNA, which indicates the importance of investigating different nanoparticles to be used in the prevention of the MPXV [70,71].

Despite the great potential of the nanomedicine field, published work is limited to in vitro and in vivo research, while their applicability in clinical practice has yet to be achieved. However, the rapid spread of the MPXV in non-endemic countries will lay the foundation for accelerating next-generation MPX vaccines, antivirals, and diagnostics to monitor and control the MPXV [59].

## 6. MPXV Effect on the Immune System

Innate immune cells typically act as the first line of defense following MPX infection, with numerous in vitro and in vivo studies demonstrating that monocytes and neutrophils are the initial targets and the predictors of MPXV lethality [72]. Following an infection and drainage to the lymph nodes (LNs), the MPXV replicates extensively in the lymphoid tissues located in the neck and throat [72]. Furthermore, data showed that the MPXV targets monocytes and macrophages, dendritic cells (DCs), B cells, and activated T cells [72]. After which, the virus disseminates along the lymphohematogenous route to distant organs, including the liver, spleen, lungs, kidney, intestines, skin, and other organs [72]. Moreover, the MPXV was shown to affect the immune system (e.g., immune cells, immune components, etc.). A study showed that despite being able to detect vaccinia virus-infected monocytes and produce inflammatory cytokines (e.g., IFNγ and TNFα), MPXV-specific CD4+ and CD8+ T cells were largely incapable of responding to autologous MPXV-infected cells.

Further analysis revealed that the MPXV does not interfere with expression or intracellular transportation. Instead, it prevents T cell receptor-mediated T cell activation in trans, triggering a state of non-responsiveness by blocking antiviral T cell activation and inflammatory cytokine production, contributing to viral dissemination in the infected host [73]. Notably, when comparing the MPXV to the variola virus, the MPXV was found to have an ortholog to COP-A44L that encoded the 346 aa proteins. In contrast, the variola virus was found to have a 210 aa fragment of its ortholog that is missing the N-terminal domain. A 3-β-hydroxysteriod dehydrogenase, encoded by the COP-A44L, functions by converting the pregnenalone to progesterone and dehydroepiandrosterone to androstendione [74]. Such a reaction is crucial for making all types of steroid hormones, including glucocorticoids, known for their immunosuppressive and anti-inflammatory activities that can affect the host’s antiviral immune response [74]. Therefore, using nanoparticles with immunomodulatory activity can target the immune cells and prevent viral interaction with the host cell, helping to develop new therapeutic and preventative strategies.

Immunotherapy is emerging as an effective preventative and treatment strategy for many diseases, including infectious diseases. Therefore, developing and studying immunomodulatory nanosystems can rapidly overcome many obstacles that are faced in the field of infectious diseases [75]. To date, various nano-formulations display important immunomodulatory capabilities. Some have immunostimulatory activities, while others have immunosuppressive activities, which could be used to control the release of immunoregulatory agents, antigens, and adjuvants. Furthermore, nanoparticles with immunomodulatory activities can be further functionalized and decorated with agents to achieve targeted delivery and boost the encapsulated agent’s efficacy. For example, decorating the surface of the nanomaterials with mannose or the CD11c, CD40, and DEC-205 antibodies can promote their internalization by DCs through receptor-mediated endocytosis [75].

Similarly, decorating nanoparticles with CD44, lectins, dextran, and folate makes them more recognizable by the corresponding receptors overexpressed on macrophages, with dextran having intrinsic targeting properties for macrophages [75]. Additionally, modifying their surface with CD3 Ab and/or a tLyp1 peptide showed increased nanoparticle uptake by T cells and regulatory T cells (Tregs), respectively. Recent studies showed that functionalizing the nanoparticles with albumin-binding domains in a process called “albumin hitchhiking” enabled the nanoparticles to drain to the LNs. These different techniques for functionalizing nanoparticles can be utilized to prevent or resist bacterial or viral infections, such as HIV, influenza, hepatitis, MPX, etc. [75]. In this aspect, nanoparticles with anti-infective vaccines and specific antigenic components are used instead of whole microbes to increase immune efficiency. Thus, protecting the antigens from early degradation, clearance, and elimination from blood circulation.

Moreover, not using the nanoparticle anti-infective vaccines often requires the assistance of adjuvants to effectively activate the immune systems. Virosome- and liposome-based nanovaccines were tested in the face of infectious diseases and showed promising efficacy in the human body. Until now, two nanoparticle-based vaccines (Epaxal and Inflexal V) have been approved by the FDA for the prevention of malaria, influenza, and hepatitis A [75]. Numerous pieces of evidence have confirmed the encouraging effects of nanoparticle-based vaccines against infectious diseases. Such evidence can be used to develop a nanoparticle-based vaccine that could prevent the spread of the MPXV while also improving the immune response, particularly with nanoparticles enhancing the delivery efficiency, carrying adjuvants, and slowly releasing loaded agents [75].

## 7. Challenges and Limitations

As with the development of other vaccines, using materials to develop nanovaccines is accompanied with many challenges and limitations, which vary depending on the type of nanomaterial used, with inorganic nanoparticles having more challenges than organic ones; and the type of vaccine carried (i.e., is it inactivated, live attenuated, or a protein subunit). Indeed, there are many studies in the literature discussing the challenges associated with the type of nanomaterial used, thereby determining their potential drawbacks, such as ensuring their biodegradability and biocompatibility, tailoring the elected immune response, and the ability to minimize their toxicity [59]. In terms of the vaccine’s interactions with the used nanomaterials, studies have shown that some nanoparticles can enhance the efficacy of the carried drug, as shown in a study conducted by Zhao et al. [76]. Other studies suggested the benefit of using LNPs as carriers for the mRNA since great advances were made in this field, with data demonstrating the potential LNPs have in delivering mRNA-based vaccines. This is because LNPs are considered a stable delivery system that can preserve mRNA integrity, maintain their cellular uptake, and enhance the efficiency of their delivery and nucleic acid release in the host cells [59]. Despite having an idea about the challenges and drawbacks associated with the different types of nanomaterials, it is important to fully investigate and study the toxicity, immune interaction, and effect of nanoparticles on the carrying agent each time a nanovaccine is designed (e.g., when designing a vaccine for the MPXV). Finally, other challenges that could be faced and need to be taken into consideration when designing a nanovaccine for the MPXV are the cost effectiveness of the vaccine, the ability to scale it up, and its stability in different environments, as many of the MPX cases are in many counties that cannot afford strict storage and transport requirements.

## 8. Conclusions

In summary, despite the great international efforts, MPX cases continue to increase, and it is now detected in 110 countries, probably because it is underestimated since many cases were and still are unreported or undiagnosed. Because the MPXV was previously confined to Central and West African countries, with minimal studies and research conducted in this area, more effort is required to develop better preventative and treatment strategies, especially with only four returns found for the nanomedicine, nanoparticles, and MPXV keywords in PubMed (as of 20 January 2023), and even fewer returns in other research websites. This indicates how limited research was in this field until recently, when the focus shifted to the MPXV when the MPX disease spread to the rest of the world. Now the US has the world’s largest MPX outbreak, with more than 6600 cases across 48 states [55]. For these reasons, we need to understand now more than ever that infectious diseases, particularly zoonotic infections and viruses, will spread to the rest of the world regardless of their geographical origins, with the COVID-19 pandemic being a dramatic example. Therefore, we should now translate the advances and knowledge achieved with other viral infections to prevent MPX infection and transmission. Such advances include using nanotechnology and nanomaterial sciences to develop non-conventional tools. The nanomedicine field is fast-growing, and the interaction of the nanoparticles with biological systems is now under investigation to benefit the most from this highly promising approach. Furthermore, this field holds great potential for triggering and developing new therapeutic, preventative, and detection strategies and enhancing the immune response to viral infections. To do so, more collaborative research is required to understand the similarity between different viruses in terms of the affected host cell, involvement of immune cells and components, and viral life cycle behavior. Developing such an understanding will lay the groundwork for the development and exchange of nanomaterials for use and functionalization against viral infections such as the MPXV.

## Figures and Tables

**Figure 1 vaccines-11-00428-f001:**
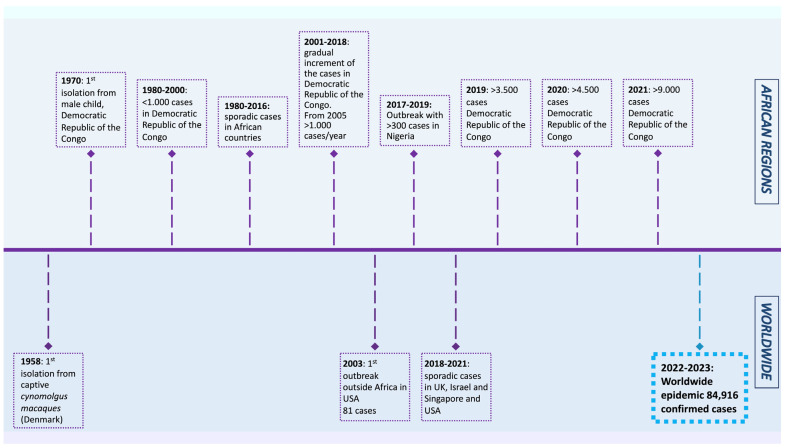
Timeline displaying the history of monkeypox outbreaks (1970–2023) [1,4,6,7,8,9,10,11,12].

**Figure 2 vaccines-11-00428-f002:**
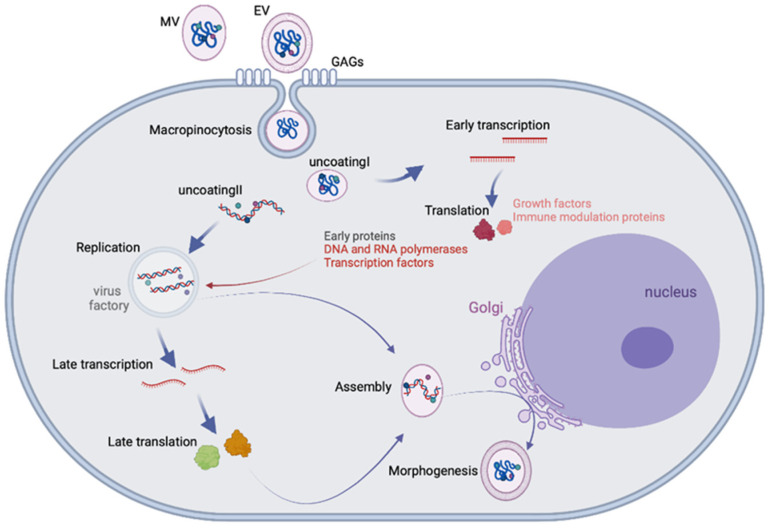
Orthopoxvirus replication cycle.

**Figure 3 vaccines-11-00428-f003:**
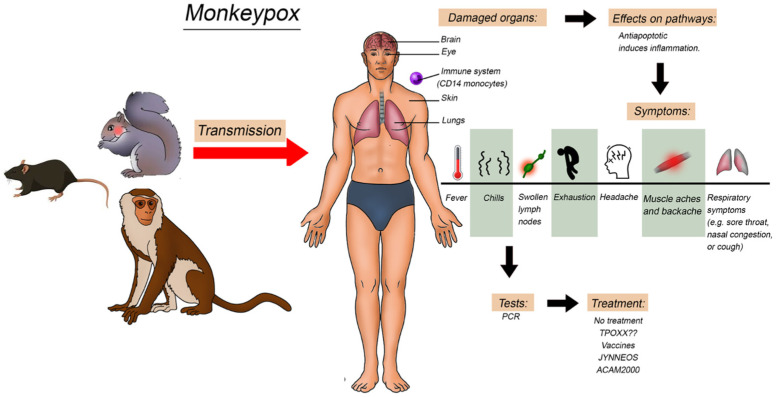
Illustration of the monkeypox transmission to humans and the effect on the different organs, symptoms, and pathways involved.

**Figure 4 vaccines-11-00428-f004:**
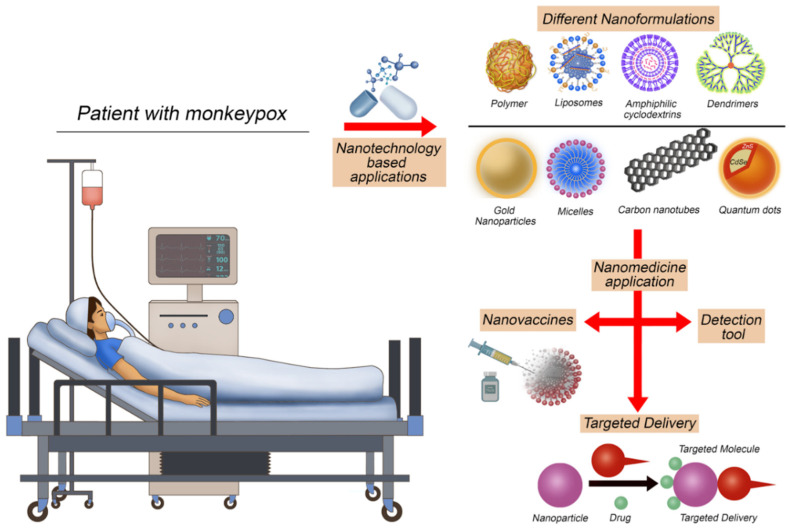
Nanotechnology applications that could be used to face the current monkeypox epidemic.

## Data Availability

Not applicable.

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
