# Peer review of "Nanomedicine as a Potential Tool against Monkeypox"

_vaccines, 2023, doi:10.3390/vaccines11020428_

Round 1

Reviewer 1 Report

This manuscript presents only two bibliographical references concerning the investigation of the use of nanoparticles in the treatment of monkeypox virus. The other 6 references in which nanoparticle results are analysed are Covid studies. Thus, the title of the article does not fit the manuscript. In my opinion, a review of the literature on the subject is not carried out, so the article should not be considered for publication.

Author Response

More studies about the nanomedicine applications in monkeypox have now been included in the manuscript. The literature regarding MPXV is very scarce, since until recently, MPXV has been confined to the African countries where it is endemic, while outside Africa the cases were rare and sporadic. In the last period, the attention towards MPXV has increased as many cases occur worldwide, with an unprecedent number of reported infections in new groups of individuals. Therefore, few studies regarding MPXV have been published so far. For this reason, in our paper, we cited COVID-19 related literature, as a lesson to discuss the potentiality of applied nanomedicine in the field of MPXV, where little has been done so far. However, new references are now inserted in the text, nevertheless, no one is actually employing this in the clinical practice (for MPXV) but there is a recent study conducted by Alec et.al. (Referenced now in the revised version of the review) where they employed a lipid nanoparticle to carry an mRNA based vaccine form MPXV. Similarly to the COVID-19 vaccines in their paper they choose the lipid nanoparticles-mRNA in developing a MPXV vaccine. Finally, applications made in nanomedicine for a specific field or disease or vaccine in our case does not mean that it is strictly limited to it, as it could be translated to other diseases and in developing other vaccines. Developing the COVID-19 vaccine was based on the mountain level of evidences and studies that were conducted to serve other diseases and viruses that were used as references, yet they are the reason why we managed to face that pandemic. Therefore we don’t see why the same can’t be done by translating the advances made in the past three years in developing a preventative strategy for MPXV.  The title of the review would highlight this issue, indeed, nanomedicine is proposed as a potential weapon against MPXV, however we decide to emphasize the concept, and the title is revised in: “Nanomedicine as a potential tool against monkeypox

Reviewer 2 Report

Manuscript titled "Nanomedicine a potential weapon against monkeypox" by Nura et al is a nice piece of writing. Authors are able to give a clean picture of monkeypox virus in a very nice an clear way. Author also highlighted major events associated with this virus. I will definitely like to see the draft published in journal. Said this i think author need to do minor polishing wit the draft. My specific comments and suggestions to authors are given below.

1) Draft need to be check for writing issues from start till end. So many extra spaces and at many places no spaces where needed.

2) In line 96, author directly used abbreviation EGFR, author need to define this.

3) There is a problem with numbering of section, most of the time numbering of sub heading is missing.

4) In line 69-70, I think author mean 10-5 and 10-6 , correct this in text

5) In a section where author discuss about vaccines and vaccination, author should also mention major issues associated with available vaccines as this become even more important in African countries which faces major Monkeypox infection. Author should include recent references which discuss major issues, concern associated with vaccines and also discuss ways to address the same. Like poor thermal stability, effect of war, uncertain natural weathers and geopolitics.

6) Term invitro and invivo should be changes to in vitro and in vivo

Author Response

Reviewer's General Comment: Manuscript titled "Nanomedicine a potential weapon against monkeypox" by Nura et al is a nice piece of writing. Authors are able to give a clean picture of monkeypox virus in a very nice an clear way. Author also highlighted major events associated with this virus. I will definitely like to see the draft published in journal. Said this i think author need to do minor polishing wit the draft. My specific comments and suggestions to authors are given below.

Answer: We thank the reviewer and the comments are now included in the revised version.

Comments:

  1. Draft need to be check for writing issues from start till end. So many extra spaces and at many places no spaces where needed.

Answer: The revised manuscript has now been checked for grammatical and space issues.

  1. In line 96, author directly used abbreviation EGFR, author need to define this.

Answer: The full name has now been added.

  1. There is a problem with numbering of section, most of the time numbering of sub heading is missing.

Answer: Sections and sub numbering has now been added to the manuscript.

  1. In line 69-70, I think author mean 10-5 and 10-6 , correct this in text

Answer: Now corrected.

  1. In a section where author discuss about vaccines and vaccination, author should also mention major issues associated with available vaccines as this become even more important in African countries which faces major Monkeypox infection. Author should include recent references which discuss major issues, concern associated with vaccines and also discuss ways to address the same. Like poor thermal stability, effect of war, uncertain natural weathers and geopolitics.

Answer: The section was carefully revised accordingly to the useful reviewer’s comments and the current available vaccines were better described.

  1. Term invitro and invivo should be changes to in vitro and in vivo

Answer: the terms are changed throughout the text

Reviewer 3 Report

It was a review paper about the application of different nanomaterials against the monkeypox virus. Here are some comments on this study that should be considered before publication:

1.     “among which the real-time PCR (RT-PCR) is the preferred routine method (Figure 1).” It seems the number of the figure mentioned here is not correct. Please check and correct it. The same in this sentence “… the vaccinia immunoglobulin as they are preferred over the brincidofovir and the cidofovir, particularly with cidofovir being a teratogen[49] (Figure 1).”.

2.     There are some grammatical mistakes in the text that should be corrected.

3.     “Similarly, other reports showed that silver nanoparticles (AgNPs; size 10–80 nm) have antiviral activity against MPXV with nanoparticles size 10 nm having more antiviral potency and reducing plaque formation [59].” please rewrite this sentence. The same for this sentence “The usage of nanoparticles will preserve the antigens as without the nanoparticles they can easily be degraded and eliminated from the blood circulation”.

4.     Conclusion part is poor. Please improve it.

5.     The nanomedicine part of the paper is so weak. You need to add more decision on the application of different types of nanomaterials and their mechanism of action for the treatment of monkeypox.

6.     Please provide the limitations related to the application of nanomedicine against monkeypox. 

Author Response

Reviewer’s General Comment: It was a review paper about the application of different nanomaterials against the monkeypox virus. Here are some comments on this study that should be considered before publication:

  1. “among which the real-time PCR (RT-PCR) is the preferred routine method (Figure 1).” It seems the number of the figure mentioned here is not correct. Please check and correct it. The same in this sentence “… the vaccinia immunoglobulin as they are preferred over the brincidofovir and the cidofovir, particularly with cidofovir being a teratogen[49] (Figure 1).”.

Answer: Now the correct figures number is added.

  1. There are some grammatical mistakes in the text that should be corrected.

Answer: Grammatical mistakes and now checked and corrected.

  1. “Similarly, other reports showed that silver nanoparticles (AgNPs; size 10–80 nm) have antiviral activity against MPXV with nanoparticles size 10 nm having more antiviral potency and reducing plaque formation [59].” please rewrite this sentence. The same for this sentence “The usage of nanoparticles will preserve the antigens as without the nanoparticles they can easily be degraded and eliminated from the blood circulation”.

Answer: Sentences are now re-written.

  1. Conclusion part is poor. Please improve it.

Answer: The conclusion section is carefully revised

  1. The nanomedicine part of the paper is so weak. You need to add more decision on the application of different types of nanomaterials and their mechanism of action for the treatment of monkeypox.

Answer: We thank the reviewer for the comment, we have now added more references about the nanomedicine applications in developing a vaccine for the Monkeypox. Regarding the different types of nanmoaterials that are out there and could be used as carrier for the Monekypox vaccine we wanted to mention that in this review we tried to put together all the studies out there that worked in developing nanovaccines for Monkeypox and since we have recently published an extensive review about that subject and we didn’t want to repeat ourselves its now cited in this review. Furthermore, in this study we focused on the nanomaterials that were shown to have either activity against viruses or have been used in developing nanovaccines which are the metal nanoparticles and the lipid nanoparticles.

  1. Please provide the limitations related to the application of nanomedicine against monkeypox. 

Answer: A section about the limitations, challenges and drawbacks the nanomedicine field has in developing a nanovaccine for the Monkeypox has now been added to the revised version.

Round 2

Reviewer 3 Report

-